# Factors Affecting Environmental Pollution for Sustainable Development Goals—Evidence from Asian Countries

Nguyen Thi Phuong Thu [ID], Le Mai Huong [ID] and Vu Ngoc Xuan *[ID]

Centre for Analysis Forecasting and Sustainable Development, National Economics University, 207 Giai Phong Road, Hai Ba Trung Dist., Hanoi 100000, Vietnam
* Correspondence: xuanvn@neu.edu.vn

**Abstract:** The world is faced with climate change and gradual increases in seawater and carbon dioxide levels, and leaders of countries all over the world need to take action in order to achieve the Sustainable Development Goals (SDGs). This paper aims to identify the factors affecting environmental pollution in Asian countries for sustainable development. This study collected data from the World Bank covering 2000–2020 for 15 Asian countries. The data were processed via STATA 17.0; the study employed the unrestricted fixed effect to solve the research problems. The empirical results suggest that electricity consumption, fossil fuel consumption, renewable consumption, population, imports, and exports affected environmental pollution in the 15 Asian countries. In addition, electricity consumption and fossil fuel consumption had a strong positive effect on Asia's environmental pollution. Moreover, population and renewable consumption negatively affected $CO_2$ emissions. These results indicate that, if an Asian country's electricity consumption increases by 1%, then its $CO_2$ emissions will increase by 0.674%; if an Asian country's fossil fuel consumption increases by 1%, then its $CO_2$ emissions will increase by 0.203%; if an Asian country's renewable consumption increases by 1%, then its $CO_2$ emissions will decrease by 0.01568%; if an Asian country's export of goods and services increases by 1%, then its $CO_2$ emissions will decrease by 0.054%; if an Asian country's import of goods and services increases by 1%, then its $CO_2$ emissions will increase by 0.067%; if an Asian country's population increases by 1%, then its $CO_2$ emissions will decrease by 0.2586%. Based on the empirical results, the study suggests new policies for green energy to achieve the Sustainable Development Goals (SDGs).

**Keywords:** electricity consumption (EC); renewable consumption (RC); population (PO); carbon dioxide emissions ($CO_2$); foreign direct investment (FDI)

## 1. Introduction

The world is faced with climate change and increases in global warming and carbon dioxide emissions. Therefore, leaders all over the world need to take action in order to reduce $CO_2$ emissions in the atmosphere. The studies regarding increases in $CO_2$ emissions and environmental pollution are carried out by scientists and policymakers. Aghasafari et al. studied the link between $CO_2$ emissions, exports, and foreign direct investment. Their empirical evidence from the Middle East and the North Africa region showed that exports and FDI affect $CO_2$ emissions [1]. Andersen et al. studied $CO_2$ emissions from the transport of China's exported goods. Their results showed that the transport of China's goods increases $CO_2$ emissions [2]. There is a research gap in studies about environmental pollution in Asian countries and how renewable consumption decreases $CO_2$ emissions. This study focuses on the study of the relationships of electricity consumption, fossil fuel consumption, renewable consumption, the import and export of goods and services, and population with environmental pollution in 15 Asian countries. These countries are China, Hong Kong, India, Indonesia, Israel, Japan, South Korea, Malaysia, Pakistan, Philippines, Qatar, Saudi Arabia, Singapore, Thailand, and Vietnam.

Concerning the role of trade, developing and developed economies have different policies. For instance, the International Monetary Fund (IMF) and the World Bank recommend developing and emerging economies to have export diversification strategies in order to reduce the dependency on specific exports and stable revenues. However, the consequence of export diversification policies might harm the environment. The export-related policies for developing and developed economies might conflict with the Sustainable Development Goals and with environmental preservation objectives. Such context leads researchers to question whether such trade strategies are good or bad for the environment and cleaner production. This is because export diversification is closely associated with energy usage and overall energy mix, while most developing and developed economies consume abundant fossil fuels and non-renewables. In recent years, climate change has become a serious threat to the developing and developed world, and it is mainly related to the use of fossil fuels and non-renewable energy sources for production, manufacturing, and urbanization processing. More precisely, the share of fossil fuels in several developing and developed countries is still dominant, contributing to an increase in greenhouse gas and carbon emissions. More specifically, there is a dearth of the literature regarding the effects of exports, imports, population, and energy consumption on environmental quality for developing and developed economies. The present study is an attempt to cover this research gap and to report the role of diversification policies and new solutions regarding climate change.

This study uses the ordinary least square (OLS) to estimate $CO_2$ emission functions by using Stata 17.0. This study aims to analyze the determinant effects of $CO_2$ emissions on the Sustainable Development Goals (SDGs). In particular, the research gap concerns the lack of use of Cobb–Douglas functions to analyze the determinants affecting environmental pollution in Asian countries. This paper also finds new information indicating that population negatively affects environmental pollution in Asian countries.

The study consists of five sections: Following Section 1, an introduction to the topic, Section 2 presents a literature review. Section 3 contains the data and methodology. Section 4 describes the data and regression analysis and presents the results of the study. Section 5 provides conclusions and suggests policies based on the results.

## 2. Literature Review

Banerjee studied the FDI flow in the energy sector among BCIM, BIMSTEC+1, and ASEAN+4 sub-regional alignments [3]. He noted that the FDI flow affected $CO_2$ emissions in these countries.

Bassey Enya et al. used the Autoregressive Distributed Lag (ARDL) model to show the link between corruption and economic growth in Nigeria [4].

Chen et al. stated that there are structural relationships among night tourism experiences, lovemarks, brand satisfaction, and brand loyalty on "Cultural Heritage Night" in South Korea [5]. This study provided some solutions for sustainable development in South Korea.

Fadly examined the green industry in Vietnam. He provided some recommendations regarding environmental management standards and sustainability resource efficiency in SMEs [6].

Destek et al. investigated the supply chain uncertainties of small-scale coffee husk-biochar production. They found that these products decrease $CO_2$ emissions in Vietnam [7].

Huang et al. studied renewable energy and $CO_2$ emissions in their paper. The empirical evidence from major energy-consuming countries showed that renewable energy decreases $CO_2$ emissions [8].

The relationship between economic growth, FDI, and environmental pollution is discussed in many papers. Le et al. showed that environmental pollution and FDI positively affected GDP in Vietnam [9]. Joo et al. also showed a positive relationship between FDI and economic growth [10]. In addition, they found that variances in host country characteristics also affected economic growth.

Chen et al. examined the sustainable development of various economies. They found that the development of the night-time economy in South Korea helped to reduce $CO_2$ emissions [5]. Khan et al. studied the innovations in energy consumption that aim to decrease carbon dioxide emissions in countries all over the world. They conducted an empirical investigation of some countries [11]. Farooq et al. also showed that the increase in border areas in an integrated region of China helped to improve sustainable development in these areas [12].

Liem et al. studied the reduction in greenhouse gas emissions from seedless lime cultivation using organic fertilizer in a province in the Vietnam Mekong Delta region [13]. Their research focused on the use of organic fertilizer to reduce $CO_2$ emissions in the north of Vietnam.

Fadly focused on the environmental management standard in Vietnam. He discussed the sustainability efficiency of Vietnamese firms that care about the green industry; that is, the necessity of reducing $CO_2$ emissions [6]. Nguyen, P.T., said that carbon emissions decrease with the value added in exports in Vietnam [14].

Nguyen (2016) also showed that regional characteristics affect the ability to attract FDI in Vietnam [15]. Nguyen provided some solutions to attract FDI for sustainable development in Vietnam [15]. Nguyen also studied financial development and the use of more renewables in Southeast Asian countries and noted the role of organic waste materials in reducing $CO_2$ emissions [16].

Fernando attempted to model FDI with the aim of helping small and medium-sized hospitality enterprises to develop sustainability [17]. Flammini, A. et al. suggested the use of activated carbon in coffee shops in Vietnam. They also discussed the introduction of green products into the circular economy [7]. Khan also found a relationship between innovations in energy consumption and $CO_2$ emissions in various countries. This research is quite new and has attracted the attention of many researchers all over the world [11].

Le et al. focused on the impacts of environmental pollution and foreign direct investment on economic growth in Vietnam using the non-linear ARDL co-integration approach. This method is quite new and helps to explain the economic growth in Vietnam in recent years [9].

Li et al. found an increase in the integration of regions in China. They noted that the sustainable development of an economy requires integration, a green economy, and reductions in carbon dioxide emissions [12].

Liem et al. showed that the use of organic fertilizer in a province in the Vietnam Mekong Delta region helped to reduce environmental pollution. They suggested that the Vietnam government should promote policies regarding the use of organic fertilizer [13]. Nguyen et al. examined the role of organic waste material in Southeast Asian countries. Policymakers need financial sponsors to obtain green waste material [16].

Gnangnon et al. studied export decisions and credit constraints under institution obstacles. They stated that these issues affect the Sustainable Development Goals. Phan noted the relationships of working conditions, export decisions, and firm constraints with sustainable development in small and medium-sized Vietnamese enterprises [18]. Phan et al. also examined the relationship of export decisions and credit constraints with the sustainable development of Vietnamese firms in recent years [19].

Raihan et al. studied the impacts of industrialization, energy consumption, and forest area on $CO_2$ emissions in Russia [20]. They suggested that carbon dioxide ($CO_2$) emissions contribute significantly to global climate change, which, in turn, threatens the environment, development, and sustainability.

Shahzad et al. also studied export product diversification and $CO_2$ emissions using contextual evidence from developing and developed economies [21]. Tsai et al. noted the impacts of environmental certificates and pollution abatement equipment on SMEs' performance. They provide empirical results for Vietnam regarding sustainable development [22].

Van Thanh studied the optimal waste-to-energy strategy by using a fuzzy MCDM model. His paper provides some solutions for sustainable solid waste management [23]. Wang et al. also showed the Sustainable Development Goals of nine provinces in China [24,25].

In summary, recent studies do not refer to the determinants affecting environmental pollution in Asian countries. The research gap concerns the new literature and empirical results regarding how the renewable consumption and population in Asian countries affect environmental pollution. In most papers on relevant topics, the scientists only focus on economic growth and how to attract foreign direct investment and create more jobs for laborers. Therefore, this paper focuses on the determinants affecting environmental pollution in Asian countries. The data and methodology used in this paper are described in the following section.

## 3. Data and Methodology

### 3.1. Data

This study collected data based on the World Bank indicator from 2000 to 2020, and they contained the following:

$CO_2$ emissions (million ton per year): this was used to measure environmental pollution in Asian countries. These data were collected based on the World Bank reports. This study assumed that the higher the $CO_2$ emissions, the higher the environmental pollution in Asian countries. Although environmental pollution can be caused by various other sources, such as waste material, industrialization, and water pollution, this paper only collected data on $CO_2$ emissions to measure environmental pollution.

Electricity consumption (TWh): this refers to the amount of electricity generated per year. This study collected annual data on electricity consumption from the World Bank report. It is very easy to collect data on electricity consumption from the secondary data in World Bank reports.

Fossil fuel consumption (TWh): this refers to the amount of fossil fuel electricity generated per year. These data demonstrate the energy consumption from fossil fuels.

Renewable consumption (%): this refers to the amount of green electricity, such as wind, water, and solar electricity, generated per year compared to the total electricity consumption.

The export of goods and services in Asian countries: this is measured in USD. It is also measured by the percentage of the GDP of each country.

The import of goods and services in Asian countries: this is measured in USD. It is also measured by the percentage of the GDP of each country. It denotes the imports of each country in Asia, measured in USD.

Population in Asian countries: this is measured by the total number of persons.

### 3.2. Methodology

The research diagram presented in Figure 1 was used in this study.

The research model is as follows:

The study has the functions Y = F(X1, X2, X3, X4, X5 . . . ), in which the variances are the following:

Y: $CO_2$ emissions, which measure environmental pollution;
X1: electricity consumption;
X2: fossil fuel consumption;
X3: renewable consumption;
X4: export of goods and services;
X5: import of goods and services;
X6: population.

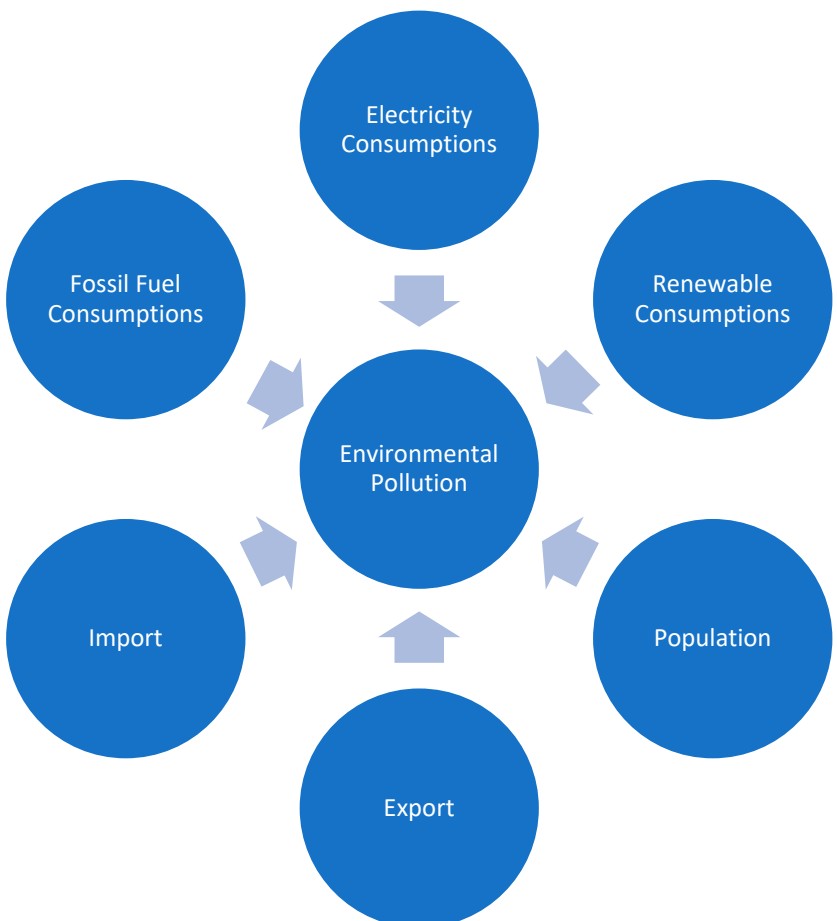

**Figure 1.** The factors linked to environmental pollution (source: compiled by authors).

This paper selects $CO_2$ emissions as the dependent variable. Our explanatory variables are electricity consumption, fossil consumption, renewable consumption, the export of goods and services, the import of goods and services, and population.

In addition, Ang and a few recent studies estimated energy consumption (or energy use) as an explanatory variable for $CO_2$ emissions by estimating an emissions–energy–output (EEO) model [25,26]. However, few of the recent studies pointed out econometric issues due to the use of energy consumption as a determinant of carbon emissions [26,27]. According to Itkonen, using energy consumption to explain $CO_2$ emissions indicates the "carbon intensity" of the fuel mix, which distorts the meaning of the parameters and exaggerates the carbon Kuznets curve (CKC) estimation [27].

However, these models can cause spurious estimations and simultaneity bias, which must be interpreted with caution. Jaforullah and King extended Itkonen's critique of the EEO model, showing that the energy variable is redundant and causes systematic volatility in the model's parameters. As a consequence, the Kuznets curve's turning point may be overestimated. Following these critiques and a few recent works [25,26], the current research mainly focuses on three approaches to draw more robust and valid conclusions. This model can be interpreted as "carbon intensity".

Following the recent literature [27,28], this study employs panel data methods to investigate the in-depth relationships between the studied factors. Panel data methods are helpful in alleviating issues related to relatively short time series, which might be useful for a better explanation of the energy–environment nexus. Our panel data allow us to treat heterogeneity problems between regions and to control for region-specific effects. Moreover, the sample is restricted to Asian countries based on the World Development Bank (2020) classification [29,30].

In summary, this paper uses the Cobb–Douglas function to estimate the independent and dependent variables by using ordinary least square (OLS). This method is very easy to apply and can be used to determine the independent variables that affect $CO_2$ emissions in the 15 Asian countries.

This study assumes the functions of $CO_2$ emissions using the Cobb–Douglas function as shown in Equation (1):

$$Y = AX1^a X2^b X3^c X4^d X5^e X6^f + e_{i,t} \tag{1}$$

As an alternative, this study can take the logarithm of both sides as shown in Equation (2):

$$Ln(Y_{i,t}) = A + aLn(X1_{i,t}) + bLn(X2_{i,t}) + cLn(X3_{i,t}) + dLn(X4_{i,t}) + eLn(X5_{i,t}) + fLn(X6_{i,t}) + e_{i,t} \tag{2}$$

In order to estimate this function to forecast the change in $CO_2$ emissions (Y), this study calculates the elasticity of Y to X1 using Equation (3):

$$Ex1 = Y' x1 \, X_1/Y = AaX1^{a-1} X2^b X3^c X4^d X5^e X6^f$$

$$X1/AX1^a X2^b X3^c X4^d X5^e X6^f \tag{3}$$

Therefore, $Ex_1$ = a.

In order to estimate this function to forecast the change in $CO_2$ emissions (Y), this study calculates the elasticity of Y to X2 using Equation (4):

$$Ex2 = Y' x2 \, X_2/Y = AbX1^a X2^{b-1} X3^c X4^d X5^e X6^f$$

$$X2/AX1^a X2^b X3^c X4^d X5^e X6^f \tag{4}$$

Therefore, $Ex_2$ = b.

In order to estimate this function to forecast the change in $CO_2$ emissions (Y), this study calculates the elasticity of Y to X3 using Equation (5):

$$Ex3 = Y' x3 \, X_3/Y = AcX1^a X2^b X3^{c-1} X4^d X5^e X6^f$$

$$X3/AX1^a X2^b X3^c X4^d X5^e X6^f \tag{5}$$

Therefore, $Ex_3$ = c.

In order to estimate this function to forecast the change in $CO_2$ emissions (Y), this study calculates the elasticity of Y to X4 using Equation (6):

$$Ex1 = Y' x4 \, X_4/Y = AdX1^a X2^b X3^c X4^{d-1} X5^e X6^f$$

$$X4/AX1^a X2^b X3^c X4^d X5^e X6^f \tag{6}$$

Therefore, $Ex_4$ = d.

In order to estimate this function to forecast the change in $CO_2$ emissions (Y), this study calculates the elasticity of Y to X5 using Equation (7):

$$Ex5 = Y' x5 \, X_5/Y = AeX1^a X2^b X3^c X4^d X5^{e-1} X6^f$$

$$X5/AX1^a X2^b X3^c X4^d X5^e X6^f \tag{7}$$

Therefore, $Ex_5$ = e.

In order to estimate this function to forecast the change in $CO_2$ emissions (Y), this study calculates the elasticity of Y to X6 using Equation (8):

$$Ex6 = Y' x6 \, X_6/Y = AfX1^a X2^b X3^c X4^d X5^e X6^{f-1}$$

$$X6/AX1^a X2^b X3^c X4^d X5^e X6^f \tag{8}$$

Therefore, $Ex_6 = f$.

This study uses a regression analysis. Using Equation (9), this study finds the following:

$$Ln(Y_{i,t}) = A + aLn(X1_{i,t}) + bLn(X2_{i,t}) + cLn(X3_{i,t}) + dLn(X4_{i,t}) + eLn(X5_{i,t}) + fLn(X6_{i,t}) + e_{i,t} \qquad (9)$$

This study assumes that the functions between the dependent variable (Y) and the independent variables X1, X2, X3, X4, X5, and X6 will be demonstrated by the Cobb–Douglas curve in the analysis, in which, the study variables can be described as follows:

Y: dependent variable representing $CO_2$ emissions; the unit is a million tons. This measures environmental pollution at a particular time.

A: a constant of $CO_2$ emissions when X1 = 0; X2 = 0; X3 = 0; X4 = 0; X5 = 0; and X6 = 0. This means that $CO_2$ emissions are measured by a million tons with the assumptions that electricity consumption equals zero, fossil fuel consumption equals zero, renewable consumption equals zero, the export of goods and services equals zero, the import of goods and services equals zero, and population equals zero.

X1: independent variable representing electricity consumption measured by total electricity consumed per year; the unit is TWh.

X2: independent variable representing fossil fuel consumption measured by total fossil fuel electricity used in each country per year; the unit is TWh.

X3: independent variable representing renewable consumption measured by the percentage of green energy to total electricity consumption in each country per year; the unit is percent.

X4: independent variable representing the export of goods and services measured by the total export product of each country per year; the unit is USD.

X5: independent variable representing the import of goods and services measured by the total import product of each country per year; the unit is USD.

X6: independent variable representing population; the unit is persons. Specifically, the following hypotheses are tested:

**Hypothesis 1 (H1):** *electricity consumption affects $CO_2$ emissions.*

**Hypothesis 2 (H2):** *fossil fuel consumption affects $CO_2$ emissions.*

**Hypothesis 3 (H3):** *renewable consumption affects $CO_2$ emissions.*

**Hypothesis 4 (H4):** *the export of goods and services affects $CO_2$ emissions.*

**Hypothesis 5 (H5):** *the import of goods and services affects $CO_2$ emissions.*

**Hypothesis 6 (H6):** *population affects $CO_2$ emissions.*

The independent variables considered in the regression model are described in Table 1.

**Table 1.** Description of the independent variables in the linear regression model.

| Variable | Explanation | Expected |
|:---:|:---:|:---:|
| X1 | Electricity consumption | + |
| X2 | Fossil fuel consumption | + |
| X3 | Renewable consumption, such as water, solar, and wind electricity consumption. | − |
| X4 | Export of goods and services | +/− |
| X5 | Import of goods and services | +/− |
| X6 | Population | +/− |

(Sources: compiled by authors).

## 4. Results

The relationships of FDI inflow and environmental pollution with economic performance have been widely discussed in recent years. However, there is a shortage of research about the impacts of renewable consumption and other factors on $CO_2$ emissions in Asian countries. Empirical studies have shown that electricity consumption increases $CO_2$ emissions. Therefore, this study researched the impacts of electricity consumption, fossil fuel consumption, renewable consumption, the export of goods and services, the import of goods and services, and population on environmental pollution. This study aimed to guide policymakers' attention toward the protection of the environment for sustainable development in Asian countries. The description analysis is presented in Table 2.

**Table 2.** Description analysis of the variables.

| Variable | Obs | Mean | Std. dev. | Min | Max |
|---|---|---|---|---|---|
| LnCO$_2$ | 315 | 5.543659 | 1.44375 | 3.410223 | 9.274993 |
| LnElectricity | 315 | 25.81675 | 1.41101 | 22.84885 | 29.6727 |
| LnFossilfuel | 315 | 27.77732 | 1.287042 | 25.58286 | 31.1444 |
| Renewable consumption | 315 | 5.428335 | 5.900787 | 0 | 24.20835 |
| LnExport | 315 | 25.97537 | 1.118355 | 23.17052 | 28.89756 |
| Lnimport | 315 | 25.83682 | 1.204807 | 22.10094 | 28.57263 |
| LnPop | 315 | 17.78745 | 1.880788 | 13.29205 | 21.06763 |

(Sources: compiled by authors based on the Word Bank data).

The data collected regarding the 15 Asian countries from 2000 to 2020 provided 315 observations. Dependent variable Y is ln of $CO_2$ emissions, with a mean of 5.543659 million tons per year, a minimum value of 3.41 million tons in 2000, and a maximum value of 9.27 million tons in 2020.

Independent variable X1 is ln of electricity consumption, with a mean of 25.81675 TWh, a minimum value of 22.84885 TWh in 2000, and a maximum value of 29.6727 TWh in 2020.

Independent variable X2 is ln of fossil fuel consumption, with a mean of 27.77732 TWh, a minimum value of 25.58286 TWh, and a maximum value of 31.1444 TWh.

Independent variable X3 is renewable consumption, with a mean of 5.428335% total electricity consumption, a minimum value of 0%, and a maximum value of 24.2%.

Independent variable X4 is ln of the export of goods and services, with a mean of USD 25.97 billion, a minimum value of USD 23.17 billion, and a maximum value of USD 28.89 billion.

Independent variable X5 is ln of the import of goods and services, with a mean of USD 25.83 billion, a minimum value of USD 21.1 billion, and a maximum value of USD 28.57 billion.

Independent variable X6 is ln of population, with a mean of 17.78745 persons, a minimum value of 13.29205 persons, and a maximum value of 21.06763 persons.

A graph of $CO_2$ emissions per capita from 2000 to 2022 in the 15 Asian countries is shown in Figure 2 in which the vertical axes are as follows:

$CO_2$ emissions per capita from 2000 to 2020 for the 15 Asian countries; the unit is tons. The horizontal axes show years from 2000 to 2020. It can be seen that Qatar has the highest $CO_2$ emissions ranging from 40 to 70 tons per capita per year. In recent years, $CO_2$ emissions in Qatar have decreased to around 40 ton per capita per year.

The country with the second highest amount of $CO_2$ emissions per capita per year at 20 tons is Saudi Arabia.

In general, the other countries have $CO_2$ emissions of around 5 to 15 tons per capita per year, and the trends remain unchanged from 2000 to 2020.

A graph of renewable consumption to total electricity consumption from 2000 to 2020 is shown in Figure 3 for the 15 Asian countries; the unit is percent. It can be seen that Vietnam has the highest percentage of renewable consumption, approximately 20%, and that the other countries have around 3–10% of renewable consumption compared to total electricity consumption.

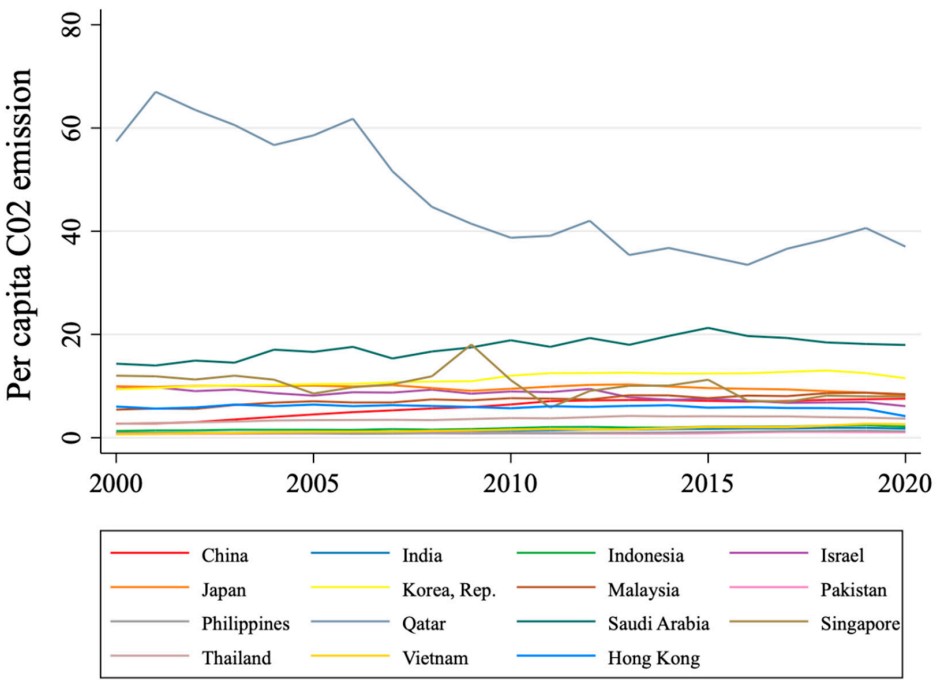

**Figure 2.** The $CO_2$ emissions per capita each year for the 15 Asian countries from 2000 to 2022 (unit: tons) (sources: compiled by authors).

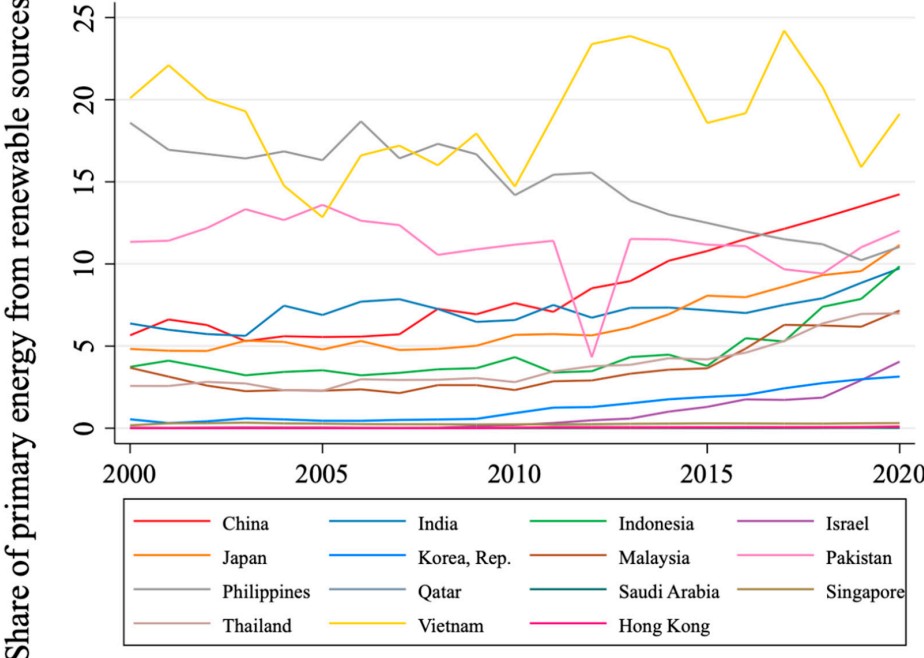

**Figure 3.** The percentage of renewable consumption to total electricity consumption from 2000 to 2020 in the 15 Asian countries (sources: compiled by authors).

The results of the regression analysis between $CO_2$ emissions (Y) and electricity consumption (X1), fossil fuel consumption (X2), renewable consumption (X3), exports (X4), imports (X5), and population (X6) are shown in Table 3. In the table, *p*-value = 0.000 shows the correlation between electricity consumption and $CO_2$ emissions.

**Table 3.** The regression between $CO_2$ emissions (Y) and electricity consumption (X1), fossil fuel consumption (X2), renewable consumption (X3), exports (X4), imports (X5), and population (X6) from 2000 to 2020 for the 15 Asian countries.

| R-Squared: | Obs Per Group: | | | | | | |
|---|---|---|---|---|---|---|---|
| Within = 0.8745 | min | = | 17 | | | | |
| Between = 0.7473 | avg | = | 20.1 | | | | |
| Overall = 0.7453 | max | = | 21 | | | | |
| | F(8278) | = | 242.11 | | | | |
| corr(u_i, Xb) = 0.4800 | Prob > F | = | 0 | | | | |
| Ln $CO_2$ emissions | Coefficient | Std. err. | t | P > t | [95% conf. | interval] | |
| Ln electricity consumption | 0.6740801 | 0.0695059 | 9.7 | 0.000 *** | 0.5372553 | 0.8109048 | |
| Ln fossil fuel consumption | 0.2037071 | 0.0879156 | 2.32 | 0.021 ** | 0.0306422 | 0.376772 | |
| Renewable consumption | −0.0160323 | 0.0037971 | −4.22 | 0.000 *** | −0.023507 | −0.0085576 | |
| Ln export of goods and services | −0.0542662 | 0.027156 | −2 | 0.047 ** | −0.1077238 | −0.0008086 | |
| Ln import of goods and services | 0.0661127 | 0.0379642 | 1.74 | 0.083 * | −0.008621 | 0.1408465 | |
| Ln population | −0.2596075 | 0.0534383 | −4.86 | 0.000 *** | −0.3648026 | −0.1544124 | |
| Constant | −11.94665 | 1.064255 | −11.23 | 0.000 *** | −14.04167 | −9.851628 | |

Note: *** $p < 0.01$, ** $p < 0.05$, * $p < 0.1$. (Sources: compiled by authors).

*p*-value = 0.021 shows the correlation between fossil fuel consumption and $CO_2$ emissions.

*p*-value = 0.000 shows the correlation between renewable consumption and $CO_2$ emissions.

*p*-value = 0.047 shows the correlation between the export of goods and services and $CO_2$ emissions.

*p*-value = 0.083 shows the correlation between the import of goods and services and $CO_2$ emissions.

*p*-value = 0.000 shows the correlation between population and $CO_2$ emissions.

Adj R-squared = 0.8745 indicates that 87.45% of the independent variables' fluctuations can explain the $CO_2$ emissions.

The results of the regression analysis show the function between Y ($CO_2$ emissions) and X1, X2, X3, X4, X5, and X6 in Equation (10) as follows:

$$LnY = -11.94665 + 0.6740801LnX1 + 0.2037071lnX2 - 0.0160323LnX3 - 0.0542662LnX4 + 0.0661127LnX5 - 0.2596075LnX6 \tag{10}$$

The results of the regression functions are very important. This study used equations to calculate the elasticity of the $CO_2$ emissions to the independent variables, namely, energy consumption, fossil fuel consumption, renewable consumption, the export of goods and services, the import of goods and services, and population. These equations help to compute the elasticity; therefore, we can forecast the fluctuations in $CO_2$ emissions for the future. These indicators can help policymakers to develop policies that protect the environment and achieve the Sustainable Development Goals (SDGs).

This study calculated the elasticity of Y to X1 as EX1 = 0.6740801.

These results indicate that, if an Asian country's electricity consumption increases by 1%, then its $CO_2$ emissions will increase by 0.674%. These values show that electricity consumption has significantly affected environmental pollution in Asian countries for a

long time. These results also indicate that the problem needs to be reconsidered. Asian countries have focused on electricity consumption for more than two decades without considering the environment.

This study calculated the elasticity of Y to X2 as $E_{X2} = 0.2037071$.

These results indicate that, if an Asian country's fossil fuel consumption increases by 1%, then its $CO_2$ emissions will increase by 0.203%. These values show that fossil fuel electricity consumption has significantly affected environmental pollution in Asian countries for a long time. These results also indicate that the problem needs to be reconsidered. Asian countries have focused on fossil fuel electricity for more than two decades without considering the environment.

This study calculated the elasticity of Y to X3 as $EX3 = -0.0160323 \times 5.42/5.54 = -0.01568$. These results show that, if an Asian country's renewable consumption increases by 1%, then its $CO_2$ emissions will decrease by 0.01568%. The empirical results show that green energy helps to reduce environmental pollution in Asian countries. These results also show that these Asian countries care about renewable energy. Asian governments already invest more into renewable energy, such as wind, water, and solar energy, to decrease environmental pollution.

This study calculated the elasticity of Y to X4 as $EX4 = -0.0542662$. This result shows that, if an Asian country increases the export of goods and services by 1%, then its $CO_2$ emissions will decrease by 0.054%. This result suggests that the export of goods and services from Asian countries to other countries, such as those in Europe and the United States of America, can decrease $CO_2$ emissions. These results also show that the export of goods and services from Asian countries to those in Europe and the United States of America in recent years has decreased environmental pollution. Customers in developed countries care about and protect the environment. Therefore, they prefer to use environmentally friendly products.

This study calculated the elasticity of Y to X5 as $EX5 = 0.0661127$. This result shows that, if an Asian country increases the import of goods and services by 1%, then its $CO_2$ emissions will increase by 0.067%. This result also shows that the import of goods and services from Asian countries increases $CO_2$ emissions. This means that the import of goods and services in Asian countries increases environmental pollution.

This paper calculated the elasticity of Y to X6 as $EX6 = -0.2596075$. This result shows that, if an Asian country's population increases by 1%, then its $CO_2$ emissions will decrease by 0.2586%. These empirical results mean that people in Asian countries have started to care more about protecting the environment in recent years. Young people are highly educated and focused on protecting the environment. Thus, the higher the population, the lower the $CO_2$ emissions.

## 5. Conclusions and Recommendations

The statistical results of the $CO_2$ emission data show that several countries have achieved stable levels of $CO_2$ emissions; however, in the 15 Asian countries, the levels of $CO_2$ showed an increasing trend from 2000 to 2020. This demonstrates that some countries are focused on and have succeeded in developing a green economy, while others have not yet successfully implemented policies for a green economy, as they have low-carbon-developed economies. Asian countries have developed their economies by attracting polluted electricity consumption and by only focusing on increasing GDP without protecting the environment.

The empirical results show the relationship between $CO_2$ emissions and electricity consumption. Therefore, the 15 Asian countries mostly consume fossil fuel energy and cause environmental pollution. The greater the electricity consumption in the 15 Asian countries is, the greater the $CO_2$ emissions are.

The nexus between fossil fuel consumption and environmental pollution is positive. The 15 Asian countries care about environmental pollution and people use more renewable consumptions. The more renewable consumptions are used, the less environmental pollution are caused. In addition, the empirical results show that the 15 Asian countries

export green products and services. The more exports they have, the less $CO_2$ emissions they have.

In other words, the import of goods and services of the 15 Asian countries causes environmental pollution. The empirical results show the relationship between $CO_2$ emissions and imports is positive. It means that the greater the import of goods and services in the 15 Asian countries is, the greater the $CO_2$ emissions are. This is because the 15 Asian countries import more products which cause $CO_2$ emissions. The 15 Asian countries' governments should have a policy to control the import of polluted products, in spite of the fact that it can be an influence in attracting FDI and economic growth.

The empirical results also show the population in the 15 Asian countries cares about environmental pollution. The greater the population is, the less $CO_2$ emissions there are. Therefore, policymakers have successfully improved the knowledge of citizens in order to protect the environment and decrease $CO_2$ emissions. Citizens in the 15 Asian countries care about the environment. They use green products and renewable products. Everybody would also like to use green energy, such as solar, wind, and water electricity.

With the levels of $CO_2$ continuously increasing in recent years in Asian countries, it was observed that the 15 Asian countries tend to import goods and services in industries that use a large amount of fossil energy (e.g., thermal power), with the technology used being obsolete. At the same time, this also indicates that the use of clean energy in the 15 Asian countries is not as effective as the use of clean energy in other countries, such as those in Europe. Given the advantages of exploiting the abundant fossil fuel resources, it is easy to fully utilize these resources. The promotion of clean energy development was, therefore, not considered urgent in the past.

To take action, everyone needs to understand and be aware of how important it is to protect the environment and reduce $CO_2$ emissions. This is the campaign of each individual, each family, each organization, each country, and the whole world.

"One tree should not be young—3 trees together make a high mountain"; a single person who protects the environment and reduces their $CO_2$ footprint will have no impact, but many people and whole societies protecting the environment will surely produce different results. Therefore, the first measure that can be taken to protect the environment is to wake up to the reality of climate change and raise the awareness of people to take action. It is not a super-invention; it is maybe only the act of using clean energy and products that do not pollute the environment.

Asian countries outperformed other regions globally in 2021 with a carbon emission reduction of 1.2% vs. 0.5% in an effort to reduce greenhouse gas emissions from economic growth despite facing many obstacles. The publication of the Net Zero Index report of countries included various information; for example, it stated that Asian countries are leading in the race to reduce carbon emissions. However, the path from the current 1.2% emission reduction to the target of 15.2% per year is still very long.

Global and national goals need to be translated into policies. The positive outcome of Asian governments is that a number of policies have been implemented. However, to maintain the 1.5 °C target, the governments in this region need to have decisive policies, including combining renewable energy targets with elimination plans, phasing out the use of coal, promoting energy efficiency with electrification policy, integrating carbon pricing policy with innovation, scaling clean technology, and ensuring a streamlined transition. The economies whose efforts align with their goals are those of Australia, China, Malaysia, New Zealand, South Korea, and, possibly, Thailand. These countries have shown encouraging progress in direction and speed. Most countries still generate a lot of carbon emissions, but they are maintaining a promising development momentum. Several economies have not experienced consistent performance and have lagged in carbon emission reductions over the past decade: India, Indonesia, and Japan. With greater ambition and determination, these countries have the ability to get back on track.

Some economies are still far from their destination: Bangladesh, the Philippines, Pakistan, and Vietnam. These developing economies started with a relatively low carbon

intensity. The economic growth of these countries over the past decade has been fueled in part by coal, and these are the biggest risks when a nation is caught between dwindling resources and climate change. While policymakers are under pressure to secure affordable energy supplies, there is still a business opportunity for net zero investments thanks to the innovation force. The increase in energy prices and the supply crisis have created a tendency to rush to fossil fuels in the short term but, at the same time, have increased investment opportunities for renewable energy in the long term.

Similarly, business drivers for energy efficiency have increased, particularly in energy-intensive sectors and industries where it is difficult to reduce energy consumption. Businesses will find ways to consume less energy and use energy more efficiently, signaling a turning point in the way we think about energy. The carbon reduction story is shifting from ambition to action. However, bridging the gap from ambition to action will be a turning point for Asia to gain a competitive edge in the global race for net zero.

This paper shows that electricity consumption, fossil fuel consumption, and the export of goods and services have a positive impact on $CO_2$ emissions. In contrast, population, renewable consumption, and the import of goods and services have a negative impact on $CO_2$ emissions. These results are similar to those obtained in the study conducted by Huang, Y. et al. [8]. These results suggest that green energy and population will have a long-term negative impact on environmental pollution in the 15 Asian countries. At the same time, this study also confirms that using a large amount of fossil fuel energy (coal, gas, etc.) or heavy industry will increase $CO_2$ emissions in the future. In addition, $CO_2$ emissions also have an impact after 1 year; in other words, after $CO_2$ is released into the environment, it will only affect the economy after 1 year, and there is no immediate impact (economies have not shown any signs of impact) of $CO_2$ emissions.

This paper has some limitations; for example, it did not examine the impacts of FDI, economic growth, innovations, and forest area in the 15 Asian countries on environmental pollution, similar to the papers produced by Le et al., Raihan et al., and Khan et al. [9,11,20] in Vietnam and Russia. This is because, in some countries, the government just cares about economic growth and attracts FDI without protecting the environment; in other Asian countries the policymaker focuses on environmental pollution. Therefore, the paper used the ordinary least square to estimate FDI, economic growth, and environmental pollution, $p$-value > 0.1 shows there is no correlation between FDI, economic growth, and $CO_2$ emissions. Future research should resolve this issue by focusing on some countries that care about environmental pollution by attracting green FDI and sustainable economic growth. In addition, future papers should focus on the countries which care about FDI and economic growth and do not care so much about protecting the environment.

This paper also lacked data on innovation and forest area to resolve these issues. In addition, the Cobb–Douglas functions and the ordinary least square (OLS) method are maybe not suitable to resolve these problems. Future research can focus on these issues in Asian countries.

**Author Contributions:** Methodology, N.T.P.T.; Software, L.M.H.; Formal analysis, L.M.H.; Data curation, L.M.H.; Writing—original draft, V.N.X.; Supervision, V.N.X. All authors have read and agreed to the published version of the manuscript.

**Funding:** This study received no external funding.

**Institutional Review Board Statement:** Not applicable.

**Informed Consent Statement:** Not applicable.

**Data Availability Statement:** Not applicable.

**Conflicts of Interest:** The authors declare no conflict of interest.

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
