# Peer review of "Factors Affecting Environmental Pollution for Sustainable Development Goals—Evidence from Asian Countries"

_sustainability, doi:10.3390/su142416775_

Round 1

Reviewer 1 Report

This paper is relatively complete and addresses a relevant theme for practice. The paper offers a set of analytical tools for factors affecting environmental pollution for SDGs. This paper aims to identify the factors affecting environmental pollution in Asia countries for sustainability development. The paper's topic and research question are relevant to sustainability journal readers. The authors explain the research questions clearly and the findings are also presented. However, I have some reservations. Most importantly, the current literature review on the factors affecting environmental pollution for SDGs in similar region is rather narrow. Some discussions should be made around the research model. In particular, the authors need to deepen empirical support and policy enlightenment on the analysis of local development initiatives environmental pollution for SDGs. Specific recommendations are as follows.

1. The abstract needs to refine the research conclusions according to the research results .

2. Research Methods. What are the problems solved by each model and the relationship between them. Analytical Model for the Cobb-Douglas as equation - Some discussions should be made around the research model, and discuss the applicability or applicable conditions of the model.

3.  Data Sources. A table description is required.

4. Figure 2. Description analysis of the variables definitely requires further development – the source is not clear. Please supply the documentary evidence of the calculation basis. The time interval should preferably be equidistant.

5. Line 340-353. What is the problem oriented CO2 emissions policy area embodied? It is suggested to add explain.

6. In conclusions and recommendations remarks, this paper has some limitations such as the authors don’t refer the impacts of FDI, economic growth, innovations and forest area. Conclusion part requires further development especially according to recommendations.

Author Response

Dear sir/madam,

Thanks again for your support. I send you the response to the comments in the attached files.

Thank you very much. You suggested the very useful comments. I think your advice make our paper more better. I hope that you feel well with my revised article.

I am looking forward to hearing from you soon.

Best regards,

Prof. Dr. Vu Ngoc Xuan

National Economics University

Reviewer 2 Report

- Title: 

Factors Affecting Environmental Pollution for Sustainability Development Goals - Evidence from the Asia Countries

The central purpose is relevant, but the paper must be improved in all their structure.

- It is necessary to avoid using the term - The authors or author. 

For example:

Line 12:  The authors collected the data based.

Line 127: The author collected data based on the World Bank indicator from 2000 to 2020.

- Literature Review: 

The authors must change the format. They can integrate with the Introduction and become the text more continuous, without broken structure, such as: 

Banerjee, K., studied...; Bassey Enya, N., H.T. James, and A. Friday Bassey, used...; Chen, N. et al., said...

- Figure 1 is not a research diagram. It represents the factors linked to Environmental Pollution.

- Data and Methodology item needs to be better described. 

- Figure 2 is not a figure.

- Figure 5 is not a figure.

- The result needs to be better presented and described. 

- The text does not present a results discussion.

Author Response

(The authors gave the same response as above.)

Reviewer 3 Report

Dear Authors,

the article raises important issues of sustainable development and environmental pollution factors.

1. Abstract should be corrected linguistically.

2. Introduction (This part must to be improved.)

The introduction is too short. It lacks a discussion and, above all, a strong presentation of the motivation to undertake research and an indication of the research gap that you are trying to fill. The article should also contain the aim of the work and a short description of the methods used. The introduction should end with the work structure (what did you do).

3. Literature review (This part must to be improved.)

The literature review as it stands is simply a collection of studies that have been undertaken in the past by researchers, and that is a good thing. However, there is no discussion with the cited studies. I miss the authors' opinions on the factors influencing environmental pollution and the idea of sustainable development. I am somewhat lacking in faith that the article will add something to the existing literature, which is a pity, because I believe that the article has potential and can help to better understand the factors of environmental pollution in the region of Asian countries.

4. Data and methodology (This part can to be improved.)

Data and Methodology explained in a clear, simple way. The study was based on aggregated data available in public comparative studies. Maybe only short time series, but sufficient for this type of research. Methodology properly described. Personally, I think the hypotheses need to be changed so that they can be clearly/simply verified. In this form, the pollutants will always have an impact (positive or negative). However, this is not a necessary change, as the table below the hypotheses shows the direction of the impact.

5. Results (This part may remain unchanged.)

Results clearly defined and interpreted. All the analyzed factors contribute to environmental pollution, which is a rather average result. Most of the assumed impact had the expected sign. It is surprising, however, that although only the hypothesis concerning the impact of population on environmental pollution was partially negatively verified by the authors. Population has an impact on the level of environmental pollution, but it is a negative impact (increase in population causes a decrease in pollution). This discovery seems to me the most important conclusion of the whole article, the least obvious one. This may indicate a further direction of research. It seems to me that you authors focused so much on the statistical interpretation of the coefficients that they forgot about the general cognitive significance of the results obtained.

6. Concusions and Recommendations (After correcting the above parts, this part may remain unchanged.)

This part is an original discussion of environmental pollution and its causes. Conclusions and comments, however, refer to information that was not previously included in the article.
In the first paragraph, the authors indicate that CO2 emissions in the 15 analyzed countries are increasing, while the reader only has data on CO2 emissions statistics (Table/Graph 2) and a downward trend in CO2 emissions per capita (Figure 3). I presume that the authors analyzed CO2 emissions in the analyzed countries, but did not address this situation in the Introduction, Literature Review or Data nad Methodology.
Similarly, in the following sections of the Conclusions and Recommendations , based on the calculated coefficients and elasticities, the reader only knows how imports of goods and services affect pollution, while the structure of imports of the analyzed countries is unknown. These data should be entered into the article, e.g. in the Introduction/Literature Review section or possibly Data and Methodology. Without it, it's hard to find in these conclusions.

Also check out figures - you should call some of them tables. I also think that the list of references is too poor (but this is not the most important correction).

I encourage you to correct the text and resubmit it to the journal. The topic is important and current.

Kind regards,

Author Response

(The authors gave the same response as above.)

Round 2

Reviewer 1 Report

The paper  has been revised and improved, but it needs to be supplemented as follows:In conclusions and recommendations remarks, this paper has some limitations such as the authors don’t refer the impacts of FDI, economic growth, innovations and forest area. Conclusion part requires further development especially according to recommendations.

Author Response

(The authors gave the same response as above.)

Reviewer 2 Report

The text still can be improved. The wording is not integrated into Literature Review and Methodology. 

And as a scientific paper, it needs a results discussion,

- The changes in Literature Review/Methodology could be improved and become the text better understood.

- The review's text has some formatting structure problems, so the editors will correct that in the publishing phase. 

- The text does not present a results discussion.

Author Response

(The authors gave the same response as above.)

Reviewer 3 Report

Dear Authors,
I am still convinced that the article raises important issues regarding sustainability and environmental pollution factors. Your correction of the article is satisfactory.
However, I still have comments that you have not taken into account or have not sufficiently corrected. However, I believe that the text has been improved enough to be published in its current version. So I leave my comments for consideration.
Parts: Abstract, Introduction, Data and Methodology, Results has been sufficiently improved.
The literature review still feels like a collection of research by scientists. You added a paragraph, but I still think that part of the scholarly discussion, the opinion of the Authors, is missing. The article is an attempt to join the existing discussion, which has great cognitive potential and can help in better understanding the factors of environmental pollution in the region of Asian countries.
Conclusions and Recommendations. I still lack basic information about the countries (CO2 emissions, population, import and export structure) that are the subject of the study - a big loss for the article.

To sum up, I encourage you to improve the text, but I leave the decision to you. Regardless of whether you take into account my suggestions so far or not, I recommend publishing the article.

Kind regards,

Author Response

(The authors gave the same response as above.)
